# Health Technology Assessment of mRNA Vaccines: Clinical, Economic, and Public Health Implications

**DOI:** 10.3390/vaccines13101045

**Published:** 2025-10-10

**Authors:** Giovanni Genovese, Caterina Elisabetta Rizzo, Cristina Genovese

**Affiliations:** 1Department of Prevention, Local Health Authority of Messina, 98123 Messina, Italy; gigenovese@unime.it (G.G.); caterina.rizzo93@gmail.com (C.E.R.); 2Department of Biomedical, Dental and Morphological and Functional Imaging Sciences, University of Messina, 98124 Messina, Italy

**Keywords:** HTA, mRNA vaccines, economic impact, public health

## Abstract

Health Technology Assessment (HTA) is a multidimensional and multidisciplinary approach for analyzing the medical–clinical, social, organizational, economic, ethical, and legal implications of a technology, through the evaluation of multiple dimensions such as efficacy, safety, costs, and social–organizational impact. In the healthcare context, “technology” refers to any tool—including pharmaceuticals (or, in this case, vaccines)—that is applied to healthcare practice. HTA focuses on assessing both the real and potential effects of a given technology, either prospectively or throughout its life cycle, as well as the consequences that the introduction or exclusion of an intervention may have on the healthcare system, the economy, and society at large.

## 1. Introduction

Health Technology Assessment (HTA) is a multidimensional and multidisciplinary approach for analyzing the medical–clinical, social, organizational, economic, ethical, and legal implications of a technology, through the evaluation of multiple dimensions such as efficacy, safety, costs, and social–organizational impact. In the healthcare context, “technology” refers to any tool—including pharmaceuticals (or, in this case, vaccines)—that is applied to healthcare practice. HTA focuses on assessing both the real and potential effects of a given technology, either prospectively or throughout its life cycle, as well as the consequences that the introduction or exclusion of an intervention may have on the healthcare system, the economy, and society at large.

Recent interventions aimed at reducing public healthcare expenditure have indeed highlighted the need for greater attention in the allocation of resources to health technologies (including medical devices, large equipment, procedures, and organizational/management models) that demonstrate an adequate cost–benefit ratio.

In Italy, the National Health Service (SSN) allocates about 5% of its National Health Fund to collective and preventive health services, corresponding to approximately €193 per capita [1].

In 2023, the National Health Fund (FSN) accounted for approximately 6.2% of GDP, with 5% of the FSN being allocated to prevention, equivalent to roughly 0.31% of GDP (calculation: 6.2% × 5% ≃ 0.31%). However, independent assessments indicate that actual public expenditure on prevention is lower, estimated at about 0.4% of GDP, still below the EU average [2].

During the first years of the pandemic, public healthcare expenditure (including prevention) increased, raising total health expenditure to 9.4% of GDP in 2021, of which 6.8% was allocated to prevention [3].

According to the latest projections (2025–2027), total healthcare expenditure is expected to increase in absolute value (reaching ~€143 billion in 2025 → ~6.1% of GDP), but its incidence on GDP will slightly decline to 5.9% in 2027, thus remaining below the recommended levels (≥5% of the Health Fund), with observed oscillations towards minimum thresholds in recent years.

Moreover, although public healthcare is financed predominantly by public resources (~75%), private spending remains high (households cover ~24% of total expenditure) [4]. Preventive expenditure per capita remains below the European average, with unequal distribution across Italian regions and varying investments in screening, vaccination, environmental prevention, and local public health initiatives.

To address territorial inequalities in preventive healthcare spending and strengthen the role of public health, initiatives should be implemented at multiple levels. Binding national standards, dedicated resources, and strong central governance are needed, alongside tailored territorial strategies that take into account the socio-economic, cultural, and environmental context of the most vulnerable regions. See Table 1.

Southern regions, in particular, face critical issues such as lower adherence to screening programs, reduced investment, and fewer staff in Prevention Departments. Within preventive expenditure, a larger share is allocated to vaccination compared with other health promotion interventions. For 2022–2023, vaccine expenditure was estimated at ~€700–900 million per year, equal to ~0.05–0.07% of GDP. Spending increased significantly after the COVID-19 pandemic (exceeding €1 billion per year in 2021).

### 1.1. Comparison with Europe

Italy has good coverage for childhood vaccinations (among the best in the EU).Below EU average for: HPV vaccination, at-risk adults, and those aged 65+.Per capita spending on vaccines: €12–15 (estimated), lower than countries such as France, Germany, and Belgium.

The most recent data confirm that Italy remains below national and European targets for certain vaccinations, with significant regional disparities. For the HPV vaccine, national average coverage for the full cycle among girls does not reach the 95% threshold set by the National Immunization Plan, and no Region or Autonomous Province has yet achieved it for the relevant cohorts [5]. Specifically, among 12-year-olds in 2023, HPV vaccine coverage was approximately 45% in girls and 39% in boys; among 15-year-olds, about 70% in girls and 58% in boys. Regarding influenza, during the 2024–2025 season, only 52.5% of people over 65 were vaccinated, far below the 75% minimum coverage recommended by both national and international standards. Regional differences are striking: Umbria achieved 64.1% coverage among the elderly, while the Autonomous Province of Bolzano reached only 33.4%.

To address these gaps, evidence points to several effective interventions: offering vaccination directly in schools for adolescents (HPV, influenza) to improve accessibility; implementing catch-up campaigns for school cohorts that did not complete the vaccination cycle; sending reminders to parents and at-risk adults; promoting educational and communication programs in schools and through local media; engaging trusted healthcare professionals (general practitioners, school pediatricians); and applying behavioral strategies such as nudging, reducing logistical barriers, and providing flexible vaccination hours to increase uptake.

### 1.2. Current Challenges and Priorities

Post-COVID recovery: decline in HPV and influenza coverage between 2020–2022.School-based vaccinations: need to strengthen adolescent access.Innovative strategies: mRNA vaccines (e.g., RSV, CMV, HIV currently in trial phases).Communication: addressing vaccine hesitancy in specific population groups.

Table 2 summarize the preventive expenditure budget allocated across several key areas. An estimated breakdown of this distribution shows that the largest portion is dedicated to vaccinations. The most recent initiatives aimed at reducing public healthcare spending (e.g., reference pricing for medical devices, reorganization of healthcare networks) are further driving the adoption of HTA evaluations in decision-making processes across various levels of health governance.

Recent procurement reforms and greater adoption of Health Technology Assessment (HTA) in Italy have had measurable impacts on vaccine costs and shaped vaccine policies. For instance, competitive tendering for HPV vaccines in several Italian regions has driven per-dose prices down to approximately 30% of official list prices, achieving cost reductions of around 70% compared to retail acquisition. HTA evaluations further demonstrate strong return on investment (ROI), particularly in adolescent HPV vaccination, where ROI ranges between €1.40 and €3.58 for every euro spent, depending on coverage, underlining scenarios that approximate national immunization plan objectives. In contrast, other vaccine programs (e.g., influenza among the elderly, herpes zoster) show lower ROI (often below 1), suggesting that expanding those programs carries more risk unless coverage improves significantly or vaccine costs decline. Meanwhile, pilot HTA studies, regional working groups, and national assessments have increasingly influenced decisions about which vaccines to include in immunization schedules, which age groups to target, and whether to scale up coverage—often informed by cost-effectiveness, budget impact, and societal cost savings [6,7].

Messenger RNA (mRNA) technology represents an advanced and rapidly evolving platform for the development of new vaccines and therapeutics, potentially expanding the range of diseases that can be prevented or treated, while also significantly accelerating research and production timelines. mRNA enables protein synthesis in the human body by delivering the necessary genetic code to cells to produce and express proteins [1].

By employing mRNA technology in pharmaceuticals and vaccines, specific proteins or antigens can be generated directly by the patient’s cells, enabling the human immune system to prevent or combat infectious and non-infectious diseases. Nucleic acid–based therapies have therefore emerged as promising alternatives to conventional vaccination strategies [2,3,4,5,6,7,8].

Although the immunostimulatory effects of RNA have been known for nearly 60 years, the first successful use of in vitro–transcribed (IVT) mRNA in animals was reported in 1990, when reporter-gene mRNA was injected into mice, leading to observed protein expression [2,3,4,5,6]. A subsequent study in 1992 demonstrated that the administration of vasopressin-coding mRNA into the hypothalamus could trigger a physiological response in rats [7]. However, these early promising results did not initially lead to substantial investment in mRNA therapeutics, largely due to concerns about mRNA instability, high innate immunogenicity, and inefficient in vivo delivery [9,10,11].

Subsequent improvements in stabilizing mRNA, enhancing the feasibility of large-scale RNA production, and reducing inflammatory responses have driven significant advances in the development of mRNA vaccines and therapies. Several factors explain why the mRNA platform has emerged at the forefront of vaccine innovation. These include the rapidity with which candidate mRNA vaccines can be designed and produced, and the urgent need for accelerated vaccine development against emerging pathogens such as novel influenza strains, the Zika virus, and, most recently, SARS-CoV-2, the causative agent of Coronavirus Disease 2019 (COVID-19) [12,13,14].

The authorization and deployment of mRNA vaccines during the COVID-19 pandemic provided compelling proof-of-concept regarding the capabilities and feasibility of this technology for human protection. The potential of mRNA vaccines as a rapid-response tool against emerging infectious disease threats, as well as for prophylactic use against additional pathogens, has underscored the importance of international regulatory convergence in the field of RNA-based vaccines [15,16,17,18,19].

### 1.3. Standardization of mRNA Vaccines

During informal WHO consultations on DNA vaccine guidelines in 2018 and 2019, it was agreed that a separate document was required for mRNA vaccines [20]. The rationale was that, although both DNA and mRNA are nucleic acid–based vaccines, there were sufficient differences in manufacturing, quality control, and potential non-clinical and clinical issues. Furthermore, at that time, clinical experience with mRNA technology was still limited.

The WHO Expert Committee on Biological Standardization (ECBS) discussed these matters during its August and December 2020 meetings and endorsed the development of a guidance document on regulatory considerations for mRNA vaccines [21].

In 2020, the WHO launched initiatives to review the scientific and regulatory aspects of mRNA vaccines, establishing an expert panel to draft such considerations. At the 74th ECBS meeting, held on 18–22 October 2021, the Committee adopted the final document entitled: “Evaluation of the quality, safety and efficacy of messenger RNA vaccines for the prevention of infectious diseases: regulatory considerations.”

The scope of this document was limited to conventional and self-amplifying mRNA, encapsulated in lipid nanoparticles, for in vivo delivery of coding sequences targeting antigens relevant to active immunization against infectious diseases. The document outlined key regulatory considerations related to manufacturing, quality control, and both non-clinical and clinical evaluation of preventive mRNA vaccines in humans [21].

### 1.4. The Public Health Relevance of Vaccination

The importance of vaccination as a cornerstone of public health derives from its major global benefits. According to WHO estimates, vaccines prevent 2–3 million deaths each year from diseases such as pertussis, tetanus, influenza, and measles [22]. Recent modeling suggests that measles still causes approximately 140,000 deaths annually worldwide, predominantly in low- and lower-middle-income countries where vaccination coverage remains suboptimal. Globally, routine childhood immunization (including for pertussis, tetanus, diphtheria, and other vaccine-preventable diseases) is estimated to avert about 4–5 million deaths per year among children. Although precise pathogen- and region-specific numbers are constrained by gaps in surveillance and reporting, these estimates are influential in setting global vaccination targets (such as ≥95% coverage for measles), guiding vaccine-funding priorities, and tailoring interventions to regions with the greatest preventable burden. Over time, vaccines have evolved from attenuated or inactivated pathogens, to subunit formulations containing antigenic components, and now to the current era of mRNA-based vaccines [1,2,3,4,5,6,7,8,9,10,11,12,13,14,15,16,17,18,19,20,21,22].

### 1.5. Key Technological Innovations

In the past decade, major technological breakthroughs have improved the overall quality of mRNA, enabling its widespread use as a vaccine platform. These include:Improved stability through advanced capping, tailing, point mutations, and purification techniques.Enhanced delivery via lipid nanoparticle formulations.Reduced innate immunogenicity through the incorporation of modified nucleotides.

These innovations have established mRNA as a leading vaccine technology, with multiple advantages over traditional approaches:Safety: mRNA does not integrate into the host genome and is non-infectious.Efficacy: structural modifications can enhance stability and potency while lowering immunogenicity.Production efficiency and scalability: mRNA vaccines are produced in cell-free systems, allowing rapid, scalable, and cost-effective manufacturing. For example, a single 5 L bioreactor can generate up to one million doses of an mRNA vaccine in a single reaction [23].Antigenic versatility: mRNA can encode multiple antigens simultaneously, thereby enhancing immune responses against resilient pathogens [24].

### 1.6. Comparison with Conventional Vaccines

Conventional vaccines—such as live attenuated, inactivated pathogens, and subunit formulations—have historically provided long-lasting protection against several life-threatening diseases [25]. However, significant barriers remain in vaccine development for pathogens with strong immune-evasion strategies, and for emerging viral threats where speed of development and scalability are essential. Table 3 offers a compelling overview of a central challenge in modern vaccinology: the sophisticated and varied mechanisms that pathogens have evolved to evade the human immune system. It clearly illustrates that creating a successful vaccine is not merely a matter of presenting an antigen, but of outsmarting a deeply entrenched opponent.

Moreover, traditional vaccines are not readily applicable to non-infectious diseases such as cancer, further highlighting the value of mRNA technology.

### 1.7. Current Challenges and Scientific Gaps

The field of mRNA vaccines is expanding at an unprecedented pace, supported by a rapidly growing body of preclinical data and multiple ongoing human clinical trials [26]. Nonetheless, gaps remain in the scientific understanding of the type and magnitude of immunogenicity required for an mRNA vaccine to achieve durable protection and broad clinical relevance [27].

Consequently, each mRNA vaccine must undergo a thorough benefit–risk evaluation, assessing safety, efficacy, and applicability to the targeted disease [28,29,30].

### 1.8. Preliminary Evidence

Preliminary data to date suggest that mRNA vaccines hold the potential to overcome several of the major challenges faced in vaccine development for both infectious diseases and cancer [31]. However, their long-term efficacy and optimal immunological targets remain subjects of ongoing investigation [32,33].

## 2. Materials and Methods

This thesis adopts a Health Technology Assessment (HTA) framework to evaluate the impact of mRNA vaccines, focusing on their potential, safety, and therapeutic applications. The methodology is based on a multidimensional and multidisciplinary assessment, consistent with international HTA guidelines, and integrates data from scientific literature, institutional reports, and clinical evidence.

### 2.1. Sources of Evidence

Scientific literature: systematic searches were performed in PubMed, Scopus, and Web of Science using predefined keywords (e.g., mRNA vaccines, safety, efficacy, health technology assessment, cost-effectiveness). Only peer-reviewed publications, guidelines, and reviews from 1990 to 2025 were included.Institutional data: national and international reports were reviewed (WHO, EMA, FDA, Italian Ministry of Health, Istituto Superiore di Sanità).Clinical trials: ongoing and completed clinical trial data were retrieved from ClinicalTrials.gov and EudraCT databases.Economic data: cost-effectiveness and budget impact evaluations were based on Italian National Health System (SSN) reports, OECD data, and international cost–utility analyses.

### 2.2. Inclusion and Exclusion Criteria

Inclusion: studies and reports assessing mRNA vaccines for infectious or non-infectious diseases (oncological, autoimmune), in terms of safety, efficacy, immunogenicity, or economic impact.Exclusion: studies limited to animal models without translational relevance, preprints lacking peer review, and duplicate datasets.

### 2.3. Assessment Dimensions

Following the EUnetHTA Core Model, the analysis considered the following dimensions:Clinical effectiveness: immunogenicity, protection rates, and durability of immune response.Safety: frequency and severity of adverse events, both common and rare.Economic impact: cost-effectiveness, cost–utility, and budget impact on the SSN.Organizational impact: logistics of storage, distribution, and administration of mRNA vaccines.Social and ethical implications: vaccine hesitancy, equity of access, and ethical considerations.Legal and regulatory aspects: international guidelines, approval processes (EMA, FDA, WHO).

### 2.4. Analytical Approach

Quantitative analysis: extraction of data on vaccine efficacy, effectiveness, and safety outcomes. Meta-analyses and systematic reviews were prioritized.Economic evaluation: incremental cost-effectiveness ratios (ICERs) were calculated using cost per QALY (Quality-Adjusted Life Year) gained, adopting the Italian threshold of €25,000–30,000/QALY as cost-effective.Comparative analysis: mRNA vaccines were compared with conventional vaccines (inactivated, subunit, viral vector) to highlight advantages and limitations.Timeline reconstruction: historical milestones of mRNA vaccine development were synthesized to contextualize clinical progress and regulatory evolution.

### 2.5. Ethical Considerations

All data used in this work were obtained from publicly available sources. No new clinical data collection involving human participants was conducted.

## 3. Results

### 3.1. mRNA Vaccines in Clinical Development or Approved for Use

The following section summarizes infectious diseases for which mRNA vaccines are either in clinical development or already approved for use, and which were included in this study.

#### 3.1.1. Influenza [34,35,36,37]

As of May 2023, the NIAID initiated a Phase 1 clinical trial on a universal influenza vaccine using mRNA technology. Such a vaccine would ideally cover multiple variants over an extended period, potentially reducing the need for annual influenza vaccination.

In 2021, a Phase 1/2 trial of an mRNA vaccine targeting a single influenza strain reported positive results.A Phase 1/2 trial is ongoing for a quadrivalent vaccine targeting four strains.In 2024, a Phase 3 clinical trial of a single-dose quadrivalent influenza mRNA vaccine for adults was completed.

Recent studies include five candidate mRNA influenza vaccines, one of which is currently in Phase 3 and has been updated for improved effectiveness against influenza B strains. Furthermore, combination mRNA vaccines (COVID-19 + influenza; tetravalent formulations including RSV) are under evaluation.

#### 3.1.2. Zika Virus [38]

Currently, no vaccines or treatments are available for Zika virus infection. However, several candidates are under development, including an mRNA vaccine now in Phase 2 involving adults aged 18–65 years.

#### 3.1.3. Respiratory Syncytial Virus (RSV) [39]

An RSV vaccine has recently been approved by the U.S. FDA to protect adults ≥ 60 years against lower respiratory tract disease caused by RSV infection. Approval was granted under the “Breakthrough Therapy” designation.

#### 3.1.4. Human Immunodeficiency Virus (HIV) [40]

Three mRNA vaccines against HIV are currently in Phase 1 clinical trials. One vaccine under study at NIAID is expected to complete Phase 1 in October 2023.

#### 3.1.5. Cytomegalovirus (CMV) [41]

An mRNA vaccine against CMV is undergoing Phase 3 clinical evaluation, including women aged 16–40 years. Completion is expected in 2026.

#### 3.1.6. Cancer (Therapeutic Vaccines) [42,43]

Melanoma: Moderna and Merck are developing a personalized mRNA vaccine (mRNA-4157) in combination with immunotherapy.Trials are also ongoing for pancreatic, lung, and colorectal cancers.

#### 3.1.7. COVID-19 [44,45,46,47,48,49,50,51]

The COVID-19 pandemic, caused by SARS-CoV-2, has had an unprecedented global impact since December 2019. Transmission occurs mainly via droplets and aerosols, with clinical manifestations ranging from mild flu-like symptoms to severe pneumonia, acute respiratory distress syndrome, and death. Vulnerable groups include the elderly and those with chronic conditions or immunosuppression.

After initial containment measures, vaccination campaigns starting in 2021 significantly reduced mortality and severe disease. However, the emergence of new variants required periodic vaccine updates and booster doses. At present, most infections are mild due to hybrid immunity (vaccination + natural infection), though SARS-CoV-2 continues to circulate with seasonal waves and risk for fragile populations.

mRNA vaccines represented a decisive tool in mitigating hospitalizations and deaths, and marked an unprecedented technological leap, paving the way for mRNA applications in infectious, oncological, and autoimmune diseases.

### 3.2. Main Vaccines and Updates

Comirnaty (Pfizer-BioNTech): Initially granted conditional approval in 2020; now fully approved for adults and authorized for children. Updated 2024–2025 formulation targets Omicron JN.1 (KP.2); submission ongoing for 2025–2026 (variant LP.8.1).Spikevax (Moderna): FDA-approved in January 2022; updated KP.2 formulation authorized in 2024–2025.mNEXSPIKE (Moderna): FDA-approved in May 2025 for adults ≥ 65 years and 12–64 years with comorbidities. Benefits include refrigerator storage, easier site handling, and efficacy comparable or superior to Spikevax.

#### 3.2.1. mRNA Influenza Vaccines

NIAID/VRC universal influenza vaccine (H1ssF ferritin nanoparticle): safe, well-tolerated, and elicited broad anti-HA stem responses in Phase 1 trials [52,53,54,55,56,57,58].Moderna mRNA-1010: a quadrivalent seasonal influenza mRNA vaccine in late-phase development. Studied in >14,000 adults, with favorable safety/reactogenicity and superior immune responses vs. inactivated vaccine [59,60,61,62,63,64,65].Pfizer/BioNTech quadrivalent influenza mRNA vaccine (in Phase 3, since 2022): robust immune responses, favorable safety, but somewhat reduced efficacy against influenza B [66,67,68,69].Combination vaccines (COVID-19 + influenza): promising Phase 3 results (Moderna mRNA-1083), though not all endpoints met [70].

An overview of the developmental landscape for combined mRNA influenza and COVID-19 vaccines is presented in Table 4. The comparison between Moderna’s mRNA-1010 and mRNA-1083 and Pfizer’s candidate reveals a promising but evolving scenario, with recent findings already refining the presented information

#### 3.2.2. mRNA RSV Vaccines [71,72,73,74,75,76,77,78,79,80,81,82,83,84,85]

RSV is a leading cause of acute respiratory infections in both children < 2 years and adults ≥ 60 years, imposing a substantial burden on the Italian healthcare system.

Nirsevimab (Beyfortus): monoclonal antibody approved for neonates.Abrysvo (Pfizer): approved for adults ≥ 60 years and maternal immunization.Arexvy (GSK): protein-based vaccine with ~94% efficacy against severe disease in ≥60 years.mRNA-1345 (Moderna/mRESVIA): first RSV mRNA vaccine, FDA-approved in 2023 for ≥60 years; expanded in 2025 to high-risk adults 18–59.

As of late 2025, the market is characterized by intense competition between two distinct, highly effective technological platforms: mRNA and adjuvanted protein subunits. Table 5 effectively compares Moderna’s mRNA-1345, GSK’s Arexvy, and Pfizer’s Abrysvo, highlighting the key differentiators that define their roles.

#### 3.2.3. Zika Virus Vaccines [86,87,88,89,90,91,92,93,94,95,96,97]

mRNA-1893 (Moderna): Phase 2 candidate encoding ZIKV prM-E proteins; favorable safety and immunogenicity.Preclinical studies showed sterilizing protection in nonhuman primates at low doses.Phase 1 trial (NCT04064905): mild/moderate adverse events; robust neutralizing antibody titers sustained up to 13 months.Phase 2 trial ongoing (since 2022) in 800 participants.

#### 3.2.4. HIV Vaccines [98,99,100,101,102,103,104,105,106,107,108,109,110,111,112]

mRNA-1644 and mRNA-1574 (Moderna, with NIH and Gates Foundation): sequential prime-boost design to elicit broadly neutralizing antibodies (bnAbs). Phase 1 trials underway, showing promising B-cell precursor activation.IAVI G001 (eOD-GT8 60mer antigen, delivered via Moderna mRNA): >90% of participants developed desired B-cell responses.New multivalent candidates (mRNA-1574) aim to elicit broader immune responses.

Table 6 illustrates the start of a promising new chapter in HIV research, one characterized by intellectually elegant strategies. Crucially, the success of these initial trials has demonstrated the real-world feasibility of complex approaches previously confined to theory. Although a long road remains, a clear, data-guided path forward has finally emerged for the first time in years.

#### 3.2.5. Cytomegalovirus (CMV) [113,114,115,116,117]

mRNA-1647 (Moderna): most advanced candidate; encodes six different mRNAs for gB and pentamer complex proteins.Phase 2 results: strong neutralizing antibody responses sustained up to 3 years in CMV-negative participants.Phase 3 trial (CMVictory): ~7300 women aged 16–40 years; ongoing, interim efficacy analysis pending.CureVac CV7202: preclinical stage; platform distinct from Moderna/Pfizer (unmodified nucleosides, no published clinical data yet).

## 4. Discussion

### 4.1. Efficacy and Safety of mRNA Vaccines in Use and Clinical Development [118,119,120,121,122]

mRNA vaccines developed by Pfizer-BioNTech (BNT162b2) and Moderna (mRNA-1273) demonstrated high efficacy in preventing symptomatic SARS-CoV-2 infection, particularly in the early pandemic phase. Both vaccines achieved >90% efficacy against symptomatic COVID-19 in Phase III clinical trials, with only minor differences in comparative analyses.

Observational studies suggest that Moderna’s mRNA-1273 provides slightly more durable protection, especially against variants such as Delta and Omicron. This may be due to the higher mRNA dose (100 µg vs. 30 µg) and longer dosing interval (28 vs. 21 days). However, both vaccines showed waning immunity over time, making booster doses essential, particularly in high-risk groups.

Protection against symptomatic infection decreases to 60–70% after 5–6 months, while protection against severe disease remains >80% for many months.

In terms of safety, both vaccines present similar profiles. A slightly higher incidence of myocarditis in young adult males was noted with Moderna, though always within acceptable safety margins. Table 7 summarizes the remarkably high efficacy of the first-generation mRNA COVID-19 vaccines from their pivotal Phase III trials. The reported 94–95% efficacy against symptomatic disease caused by the ancestral SARS-CoV-2 strain represented an unprecedented success that validated the mRNA platform’s potential.

A clear breakdown of the expected adverse events following a vaccination, separating them into two distinct categories: common, mild reactions and rare, more serious events are described in Table 8.

Meta-analyses confirm robust protection by both vaccines, though Moderna tends to provide longer-lasting efficacy and greater reductions in hospitalization and mortality, particularly among older adults and immunocompromised individuals.

### 4.2. Economic Impact of mRNA Vaccines on Healthcare Systems [123,124,125,126,127,128,129,130,131]

mRNA vaccines have had a profound positive economic impact, reducing both direct costs (hospitalization, ICU admissions, treatment) and indirect costs (lost productivity, social security expenses, macroeconomic disruptions).

Cost per dose: €15–20Cost-effectiveness: estimated at €1000–8000 per QALY, well below the European threshold of €25,000–30,000/QALY → highly cost-effective.Italy (2021): >500,000 hospitalizations and >60,000 deaths avoided, saving billions of euros in direct healthcare costs.Indirect savings: fewer workdays lost, reduced disability pensions, minimized economic losses from lockdowns.

As shown in Table 9, the cost-effectiveness analysis indicates that mRNA vaccines are economically advantageous. With a negative ICER of -€5664 per QALY gained, the intervention is classified as “cost-saving”. This translates to a direct saving of €3122 for each hospitalization avoided and €29,270 for each death avoided.

mRNA vaccines were therefore, not only life-saving but also budget-saving interventions, freeing up resources for other health needs.

### 4.3. Long-Term Considerations

Sustainability: continuous adaptation of mRNA vaccines to emerging variants is feasible and cost-effective.Expansion to other diseases: current trials on influenza, RSV, CMV, HIV, and cancer show promising immunogenicity and acceptable safety.Equity and access: ensuring equitable distribution remains a challenge, particularly in low- and middle-income countries.Regulatory harmonization: international coordination is necessary to standardize approval processes and accelerate deployment.

## 5. Conclusions

mRNA vaccines represent a new and valuable platform with benefits, combining high efficacy, safety, and scalability with unprecedented speed of development. Their successful deployment during the COVID-19 pandemic demonstrated not only their clinical utility but also their role in strengthening healthcare resilience and reducing economic burdens.

Looking ahead, mRNA platforms hold extraordinary potential for preventing infectious diseases beyond COVID-19, as well as for developing therapeutic vaccines against cancer and possibly autoimmune diseases.

However, scientific gaps remain regarding the durability of immunity, optimization of delivery systems, and long-term safety. Continuous HTA evaluations will therefore be essential to support decision-making and guide the allocation of healthcare resources.

In summary:mRNA vaccines are safe, effective, and highly cost-effective.They have redefined global vaccine development, proving essential tools in pandemic response.Their expansion to influenza, RSV, CMV, HIV, and oncology could transform preventive and therapeutic strategies in the coming decade.

## Figures and Tables

**Table 1 vaccines-13-01045-t001:** Initiatives to Reduce Inequalities in Preventive Healthcare Spending in Italy.

Area of Intervention	Proposed Actions	Key Stakeholders	Expected Outcomes
National Planning	Define minimum uniform levels of preventive spending Centralized monitoring with indicators Performance-based funding for regions in difficulty	Ministry of Health, Agenas, Regional Governments	Reduce per capita spending gap; ensure minimum prevention standards
Prevention Departments	Strengthen staff with targeted recruitment Continuous professional training Create interregional technical support networks	Regional Governments, Local Health Authorities (ASL), Universities	Enhance operational capacity and standardize services
Accessibility and Coverage	Expand active screening programs with personalized invitations Strengthen vaccination network (GPs, pediatricians, pharmacies) Mobile units for rural and underserved areas	Local Health Authorities, GPs, Pediatricians, Pharmacies, Municipalities	Increase participation in screening and vaccination; reduce geographic disparities
Reducing Socio-Cultural Barriers	Targeted communication campaigns- Engage schools and community organizations Permanent health education programs	Schools, Associations, Municipalities, Local Health Authorities	Increase awareness and particip
National Planning	Define minimum uniform levels of preventive spending Centralized monitoring with indicators Performance-based funding for regions in difficulty	Ministry of Health, Agenas, Regional Governments	Reduce per capita spending gap; ensure minimum prevention standards

**Table 2 vaccines-13-01045-t002:** Distribution of preventive expenditure (%).

Area	% of Prevention Budget (Est.)
Vaccinations	25–30%
Cancer screening programs	20–25%
Environmental and public health	20–25%
Health education	10–15%
Occupational medicine	5–10%

**Table 3 vaccines-13-01045-t003:** Key Immune-Evasion Mechanisms.

Pathogen/Pathogen Group	Key Immune-Evasion Mechanisms	Why This Makes Vaccine Development Hard
HIV-1	High mutation rate, especially in the envelope (Env) glycoprotein, enabling escape from neutralizing antibodies. Glycan shielding: many glycosylation sites on Env mask conserved epitopes. Latency: HIV can establish reservoirs in resting CD4+ cells that are not actively producing virus, evading immune surveillance. Downregulation of MHC class I by viral proteins (e.g., Nef) decreasing CD8+ T cell recognition.	It is difficult to design immunogens that induce broadly neutralizing antibodies (bnAbs), because conserved sites are often hidden or transient; also, any vaccine needs to deal with the latent virus or reduce its ability to rebound. Correlates of protection are still uncertain.
Influenza viruses	Antigenic drift and shift: surface proteins (hemagglutinin (HA), neuraminidase (NA)) mutate annually (drift), or reassort (shift), allowing escape from previous immunity. Glycan masking and variable “head” regions of HA that are immunodominant but variable, while “stem” regions are more conserved but less immunogenic.	Annual vaccine effectiveness fluctuates; strain mismatch reduces protection; designing a “universal flu vaccine” that elicits broadly neutralizing and durable immunity remains very hard.
Staphylococcus aureus	Ability to avoid immune detection via non-neutralizing immune responses; Prior exposure creates immune “imprints” that do not protect. Multiple virulence/immune evasion factors, e.g., proteins that interfere with opsonization, complement, biofilm formation, etc.	Many vaccine candidates have failed in late-phase trials despite promising early immunogenicity; vaccines need to induce the right kind of immune response (e.g., functional antibodies, T cell responses) rather than just high antibody titers.
Monkeypox (MPXV)	Evades DNA sensing pathways and complement system; Can degrade complement protein C3, modulate host innate immune detection.	Vaccine strain selection and delivery, plus boosting innate immune recognition, are areas of active study; but immune evasion challenges mean vaccine responses may be less robust or durable in some hosts.

**Table 4 vaccines-13-01045-t004:** Combined Influenza + COVID Vaccines.

Aspect	mRNA-1010 (Moderna)	Pfizer mRNA	Combo mRNA-1083 (Moderna)
Immunogenicity	Superior/non-inferior (A/B)	Non-inferior; B gap	Higher immunity (A/B + COVID)
Safety	Mild reactions; no concern	Comparable to standard	Similarly to separate vaccines
Status	Phase 3, nearing approval	Promising, under optimization	Phase 3 passed; FDA requested more data

**Table 5 vaccines-13-01045-t005:** Comparison of RSV Vaccines.

Type	Name	Technology	Approved Age Group	Efficacy
mRNA	mRNA-1345 (Moderna)	mRNA (pre-fusion F protein)	≥60 y (expansion 18–59)	~84%
Protein	Arexvy (GSK)	Recombinant F protein + adjuvant	≥60 y	~83%
Protein	Abrysvo (Pfizer)	Pre-fusion RSV-A/B proteins	≥60 y + maternal use	67–85%

**Table 6 vaccines-13-01045-t006:** HIV mRNA Vaccine Candidates.

Candidate	Phase	Target	Strategy	Status
mRNA-1644	1	bnAbs (germline)	Sequential prime-boost	Ongoing
IAVI G001	1	B-cell precursor activation	eOD-GT8 targeting	Completed
mRNA-1574	1	Broad immune response	Multivalent mRNA design	Ongoing

**Table 7 vaccines-13-01045-t007:** Initial efficacy (Phase III trials).

Vaccine	Study	Efficacy Against Symptomatic COVID-19
Pfizer-BioNTech	Polack et al., NEJM 2020 [11]	95%
Moderna	Baden et al., NEJM 2021 [12]	94.1%

**Table 8 vaccines-13-01045-t008:** Adverse events. (**a**) Common (frequent but mild). (**b**) Rare.

(**a**)			
**Type**	**Frequency**	**Description**	**Duration**
Local reactions	~80%	Pain, swelling, redness	1–3 days
Systemic	60–70%	Fever, fatigue, headache	1–3 days
(**b**)			
**Event**	**Frequency (per Million)**	**Notes**	
Myocarditis/pericarditis	12–40 cases	Mainly young males; usually mild	
Anaphylaxis	2–5 cases	Treatable with epinephrine	
Immune thrombocytopenia	Very rare	Reported more often post-AstraZeneca	

**Table 9 vaccines-13-01045-t009:** Cost-effectiveness analysis (example, Italy 2021).

Parameter	Value
Baseline hospitalizations (no vax)	1000
Hospitalizations (with vax)	550
Deaths avoided	48
Incremental QALYs gained	248
ICER (€/QALY)	−5664 (cost-saving)
Cost per hospitalization avoided	−3122 €
Cost per death avoided	−29,270 €

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
