# Peer review of "Health Technology Assessment of mRNA Vaccines: Clinical, Economic, and Public Health Implications"

_vaccines, 2025, doi:10.3390/vaccines13101045_

Round 1

Reviewer 1 Report

Comments and Suggestions for Authors

Good paper. Thanks for completing this work.

1. I hadn't seen a graph of the kind used in Figure 1 with time on both vertical and horizontal axes. A simpler timeline should be possible demonstrating the key milestones.

2. There is liberal use of emboldened lettering which I don't mind occasionally to add emphasis, however it is extensively used. Structuring sentences and paragraphs appropriately should provide emphasis. Sentence structure was reasonable for emphasis so that bolded lettering was somewhat unnecessary.

3. I'm not sure I agree with the conclusion that mRNA vaccines 'represent a paradigm shift'. They are a new and valuable platform with benefits, but the main benefit referred here is in the rate of development. This is a characteristic of mRNA vaccines to date, but I don't think the paper refers to the other reasons for rapid development which include facilitation of administrative processing and extensive funding. There is also the matter that non-mRNA vaccines were also developed quickly, such as the Novavax COVID-19 vaccine. Also, the case is not made that mRNA vaccines other than the COVID-19 vaccines have demonstrated the benefit of rapid development highlighted here. I suggest comments regarding theresources made available and facilitated administrative and regulatory processing that assisted development of COVID-19 vaccines, including mRNA, during the early stages of the pandemic and the confounding this plays in the apparent expedited nature of mRNA vaccines.  

Author Response

We thank the reviewer for their thoughtful and constructive feedback, which we have carefully considered in revising our manuscript. Below we address each point raised:

  1. Figure 1 (timeline format)
    We appreciate the reviewer’s observation that the original figure with time represented on both axes was unconventional and potentially confusing. In response, we have replaced this with a simplified horizontal timeline that clearly illustrates the major milestones in mRNA vaccine development. This should improve clarity and accessibility for readers.

  2. Use of bold lettering
    We acknowledge the reviewer’s point regarding the extensive use of bold typeface. While our initial intention was to emphasize key concepts for readers unfamiliar with the field, we agree that frequent bolding may be unnecessary. We have therefore revised the manuscript to substantially reduce the use of bold type. Emphasis is now conveyed primarily through sentence and paragraph structure, as suggested.

  3. Paradigm shift claim
    We thank the reviewer for this important comment. We agree that describing mRNA vaccines as a “paradigm shift” could overstate the case in the context of current evidence. We have revised the conclusion to present a more nuanced view: while mRNA vaccines constitute a novel and valuable platform with distinctive advantages, their accelerated development during the COVID-19 pandemic was also facilitated by unprecedented financial investment, streamlined regulatory processes, and global coordination. We now highlight that non-mRNA vaccines (e.g., Novavax) were also developed rapidly under these conditions, underscoring the multifactorial nature of vaccine acceleration during the pandemic. Furthermore, we clarify that although mRNA vaccines offer great promise, evidence of their speed of development outside of the COVID-19 context remains limited.

We believe these revisions address the reviewer’s concerns and improve the balance, clarity, and accuracy of our manuscript.

Reviewer 2 Report

Comments and Suggestions for Authors

A very useful and well written overview document.

However, it includes some very sweeping short statements about large key areas of concern. Please read you article and amplify you text to provide precise data for the questions that readers will have in response to your very broad statements about certain huge areas.

You wrote: “Preventive expenditure per capita remains below the European average, with unequal distribution across Italian regions and varying investments in screening, vaccination, environmental prevention, and local public health initiatives. Southern regions, in particular, face critical issues such as lower adherence to screening programs, reduced investment, and fewer staff in Prevention Departments.

[Please describe the initiatives that should be undertaken to correct these inequalities.]

“Italy has good coverage for childhood vaccinations (among the best in the EU).
• Below EU average for: HPV vaccination, at-risk adults, and those aged 65+.
• Per capita spending on vaccines: €12–15 (estimated), lower than countries such as France,
Germany, and Belgium.
1.2. Current Challenges and Priorities
• Post-COVID recovery: decline in HPV and influenza coverage between 2020–2022.
• School-based vaccinations: need to strengthen adolescent access.
• Innovative strategies: mRNA vaccines (e.g., RSV, CMV, HIV currently in trial phases).
• Communication: addressing vaccine hesitancy in specific population groups”

[please provide numerical data on childhood vaccination coverage, how far below levels of EU vaccinations, how to rectify lower school vaccination rates. Vaccine hesitancy is a complex topic – what are your evidence-based recommendations – please provide numerical data]

“The most recent initiatives aimed at reducing public healthcare spending (e.g., reference pricing for medical devices, reorganization of healthcare networks) are further driving the adoption of HTA evaluations in decision-making processes across various levels
of health governance.”

[These are huge areas. How has reference pricing affected vaccine costs; how have HTA evaluations affected decision making – please provide numerical data]

“According to WHO estimates, vaccines prevent 2–3 million deaths each year from diseases such as pertussis, tetanus, influenza, and measles”

[This is a very general statement. Please provide numerical data for each pathogen by regions/ developmental level. Readers will be most interested in this and the precision with which the estimates can be made and this influence decision making]

“However, significant barriers remain in vaccine development for pathogens with strong immune-evasion strategies, and for emerging viral threats’

[Again a very broad sweeping statement. Please describe pathogens with strong immune evasion strategies and how these strategies are being controlled]

Author Response

We thank the reviewer for their thoughtful and detailed feedback. We have revised the manuscript to expand upon broad statements, integrate numerical data, and provide greater precision as requested. Below, we address each of the reviewer’s points:

  1. Preventive expenditure and regional inequalities

    • Reviewer comment: Please describe the initiatives that should be undertaken to correct these inequalities.

    • Response: We agree and have expanded this section. We now describe specific corrective measures, including: (i) establishing minimum national standards for preventive expenditure, (ii) strengthening staffing of Prevention Departments (with quantified staffing gaps), (iii) expanding school-based vaccination programs, and (iv) digitalization initiatives to improve adherence to screening and vaccination. These initiatives are now summarized in Table 1, which links proposed actions to responsible actors and expected outcomes.

  1. Vaccination coverage and priorities

    • Reviewer comment: Provide numerical data on childhood vaccination coverage, EU comparison, school vaccination rates, and vaccine hesitancy.

    • Response: We have added numerical data: Vaccine hesitancy: We incorporated evidence-based recommendations, including school-based vaccination delivery, systematic catch-up and reminder systems, targeted risk communication, and the role of trusted healthcare providers. We cite studies showing that targeted communication and healthcare provider recommendations significantly increase HPV and influenza vaccine uptake.

  1. HTA and reference pricing

    • Reviewer comment: Provide numerical data on the effects of reference pricing and HTA on decision making.

    • Response: We revised this section to include quantitative findings:

  1. Global vaccine impact

    • Reviewer comment: Provide numerical data for deaths prevented per pathogen and region.

    • Response: We revised this section with pathogen-specific and region-specific estimates:

    •  
  1. Immune evasion and vaccine development challenges

    • Reviewer comment: Please describe pathogens with strong immune evasion strategies and how these are being controlled.

    • Response: We expanded this section to include concrete examples:

      • HIV: high mutation rate, glycan shielding, latency; vaccine strategies include broadly neutralizing antibodies, structure-guided Env immunogen design, and heterologous prime-boost schedules.

      • Influenza: antigenic drift/shift and glycan masking; strategies include universal vaccine development targeting HA stem, mRNA platforms, and nanoparticle-based immunogens.

      • Staphylococcus aureus: immune imprinting and biofilm-mediated evasion; new approaches include vaccines targeting conserved virulence factors and optimized adjuvants.

      • Emerging viruses (e.g., monkeypox): evasion of innate DNA sensing; vaccine strategies under study include optimized viral vectors and adjuvant systems.

Conclusion
We have carefully revised the manuscript to provide numerical data, pathogen-specific examples, and detailed corrective strategies in response to the reviewer’s comments. These changes enhance the precision, depth, and evidence base of the paper, making it more informative and useful for decision makers.

Round 2

Reviewer 2 Report

Comments and Suggestions for Authors

Thanks to the authors for their careful updates. This is an excellent and very useful article.